# PERMUTATION-INVARIANT HIERARCHICAL REPRESENTATION LEARNING FOR REINFORCEMENT-GUIDED FEATURE TRANSFORMATION

## ABSTRACT

Feature transformation aims to refine tabular feature spaces by mathematically transforming existing features into more predictive representations. Recent advances leverage generative intelligence to encode transformation knowledge into continuous embedding spaces, facilitating the exploration of superior feature transformation sequences. However, such methods face three critical limitations: 1) Neglecting hierarchical relationships between low-level features, mathematical operations and the resulting high-level feature abstractions, causing incomplete representations of the transformation process; 2) Incorrectly encoding transformation sequences as order-sensitive, introducing unnecessary biases into the learned continuous embedding space; 3) Relying on gradient-based search methods under the assumption of embedding space convexity, making these methods susceptible to being trapped in local optima. To address these limitations, we propose a novel framework consisting of two key components. First, we introduce a permutation-invariant hierarchical modeling module that explicitly captures hierarchical interactions from low-level features and operations to high-level feature abstractions. Within this module, an self-attention pooling mechanism ensures permutation invariance of the learned embedding space, aligning generated feature abstractions directly with predictive performance. Second, we develop a policy-guided multi-objective search strategy using reinforcement learning (RL) to effectively explore the embedding space. We select locally optimal search seeds from empirical data based on model performance, then simultaneously optimize predictive accuracy and minimize transformation sequence length starting from these seeds. Finally, extensive experiments are conducted to evaluate the effectiveness,efficiency and robustness of our framework. Our code and data are publicly accessible [1].

## 1 INTRODUCTION

Feature transformation aims to improve tabular feature spaces by mathematically deriving more predictive representations from original features. Although deep learning has recently achieved remarkable success, it struggles to deliver strong performance on tabular data due to heterogeneous feature types, varying feature scales, and the presence of missing values or outliers. Furthermore, the black-box nature of deep learning models limits their interpretability, hindering their application in further data analysis and critical decision-making systems. Thus, automated feature transformation, which aims to generate distinguishable and informative features to enhance predictive performance, has emerged as a significant research direction in tabular data analysis.

Existing methods for automated feature transformation can be categorized into three main groups: 1) Expansion-reduction approaches (Kanter & Veeramachaneni, 2015; Horn et al., 2019b; Khurana et al., 2016b), which first enlarge the original feature space using mathematical transformations and subsequently reduce dimensionality through feature selection; 2) Evolution-evaluation approaches (Wang et al., 2022; Khurana et al., 2018a; Tran et al., 2016), which utilize Evolutionary Algorithms (EA) or Reinforcement Learning (RL) to iteratively generate and evaluate transformation candidates based on model performance feedback; 3) Auto ML-based approaches (Chen et al.,

---

[1]https://anonymous.4open.science/r/PHER-32A6

2019a; Zhu et al., 2022b), which formulate feature transformation as a search problem, employing automated machine learning techniques to identify optimal transformations. Despite achieving considerable success, these methods still face challenges in effectively modeling intricate patterns inherent in feature transformation knowledge.

Inspired by the success of generative AI, recent advances formulate automated feature transformation as a token generation task (Wang et al., 2023; Ying et al., 2024b;a). They compress feature transformation knowledge into a continuous embedding space and subsequently explore this space via gradient-based methods to identify superior feature transformation sequences. However, there are three primary limitations: 1) **Neglecting hierarchical relationships between original features, mathematical operations, and generated feature abstractions.**

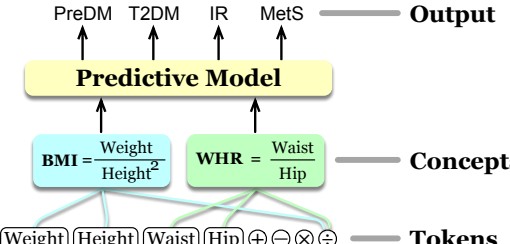

For example, as illustrated in Figure 1, Body Mass Index (BMI) and Waist-to-Hip Ratio (WHR) are high-level feature abstractions (i.e. concepts) derived from original features through mathematical transformations, significantly contributing to accurate predictions. Neglecting explicit modeling of these hierarchical relationships results in incomplete feature representations, limiting the capture of meaningful interactions and reducing the distinguishability of the embedding space. 2) **Ignoring the order invariance of the generated feature abstractions with respect to the predictive performance of the associated feature space.** Prior studies treat the entire feature transformation process as a sequential model to capture trans-

Figure 1: Illustration of hierarchical relationships between generated feature abstractions (concepts) and original features. Body Mass Index (BMI) and Waist-to-Hip Ratio (WHR) are obtained by mathematically transforming original features.

formation knowledge. However, this formulation introduces unnecessary permutation bias among generated concepts into the embedding space, hindering effective exploration and limiting the identification of the globally optimal feature transformation embedding. 3) **Relying on gradient-based search methods that strongly assume convexity of the learned embedding space.** Existing methods typically assume convexity in the learned embedding space, relying on gradient-based search methods to identify globally optimal solutions. However, in practice, due to complex interactions among features and mathematical operations, it is challenging to guarantee the convexity of the embedding space. This misalignment between the convexity assumption and the practical characteristics of the embedding space increases the risk of becoming trapped in local optima, resulting in suboptimal feature transformation sequences.

**Our Contribution: A Hierarchical Modeling and Policy-Guided Feature Transformation Perspective.** To address these limitations, we propose PHER, a novel feature transformation framework that integrates permutation-invariant hierarchical modeling and multi-objective policy-guided search. Specifically, given a large volume of feature transformation records—each consisting of a transformation sequence and the associated model performance—we first design a hierarchical modeling module. This module preserves transformation knowledge at both low-level feature interactions and high-level feature abstractions (i.e. *concepts*) into a continuous embedding space. To ensure permutation invariance, we develop a self-attention pooling mechanism that symmetrically computes attention scores across various generated concepts. This structure guarantees that any permutation of generated concepts yields identical embeddings, resulting in a global continuous embedding space that unbiasedly preserves feature transformation knowledge. Subsequently, we employ a policy-guided multi-objective search strategy to explore the global embedding space and identify the optimal feature transformation sequence. In detail, we first select the top-K feature transformation sequences based on model performance as search seeds. Then, we convert these seeds into embeddings, which serve as initial positions for exploration within the learned global embedding space. Next, we treat seed embeddings as states and implement a reinforcement learning (RL) agent to explore the embedding space, guided by maximizing downstream task performance and minimizing the length of transformation sequences. The exploratory nature of reinforcement learning enables effective navigation of the embedding space, reducing the risk of becoming trapped in local optima, even when the embedding space lacks convexity. Finally, extensive experiments are conducted on 19 real-world datasets to evaluate the efficiency, resilience, and traceability of PHER.

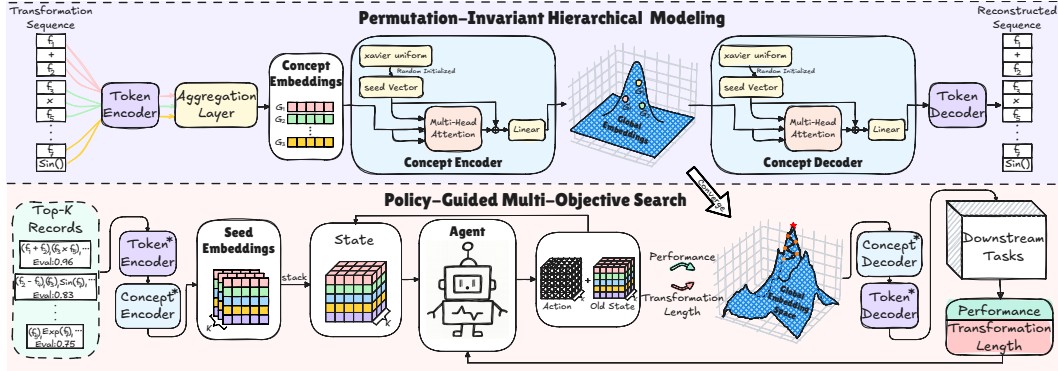

Figure 2: An overview of our framework. PHER comprises two main components: 1) Permutation-Invariant Hierarchical Modeling, which unbiasedly preserves feature transformation knowledge at both the feature-operation token level and the generated-concept level within a global embedding space; 2) Policy-Guided Multi-objective Search, which effectively explores the learned embedding space to identify optimal feature transformation sequences.

## 2 PROBLEM STATEMENT

We aim to develop an automated feature transformation framework from a generative intelligence perspective, integrating permutation-invariant hierarchical modeling and policy-guided multi-objective search strategy. Formally, given a dataset $D = \{X, y\}$, where $X$ denotes features and $\mathbf{y}$ represents the corresponding labels, along with a mathematical operation set $\mathcal{O}$ (e.g., addition, subtraction, multiplication). We first gather $n$ feature transformation records, denoted by $\{(\mathbf{\Gamma_i}, v_i)\}_{i=1}^{n}$ based on the dataset $D$, where each record consists of transformation sequence $\mathbf{\Gamma_i}$ and the associated downstream predictive performance $v_i$. We then train a token-level encoder $\phi_{tok}$ and decoder $\psi_{tok}$ to encode interactions between original features and mathematical operations into token embeddings $\mathbf{E}$, optimized through a token-level reconstruction loss. These token embeddings are grouped and averaged through an aggregation layer to form concept embeddings $\mathbf{G}$. A concept-level permutation-invariant encoder $\phi_{con}$ is then employed to eliminate order sensitivity among concepts, resulting in a global, unbiased embedding space $\mathbb{G}$. A corresponding concept-level decoder $\psi_{con}$ is simultaneously trained to capture patterns from generated concepts. After that, we employ a policy-guided strategy to explore $\mathbb{G}$, aiming to identify the optimal global embedding $\mathbf{G}'_{opt}$. This embedding can be decoded by the concept-level decoder $\psi_{con}$ and token-level decoder $\psi_{tok}$ to reconstruct the optimal feature transformation sequence $\mathbf{\Gamma}^*$, thus generating a feature space that maximizes downstream task performance $\mathcal{M}$ (See Appendix A for all notations). The optimization objective is:

$$\mathbf{\Gamma}^* = \psi_{tok}(\psi_{con}(\mathbf{G}'_{opt})) = \operatorname{argmax}_{\mathbf{G}' \in \mathbb{G}} \mathcal{M}(X[\psi_{tok}(\psi_{con}(\mathbf{G}'))]), \quad (1)$$

## 3 METHODOLOGY

### 3.1 FRAMEWORK OVERVIEW

Figure 2 illustrates the overall structure of PHER, comprising two primary components: 1) Permutation-Invariant Hierarchical Modeling and 2) Policy-Guided Multi-Objective Search. Specifically, given a large collection of feature transformation records, each record consists of one feature transformation sequence and the corresponding model performance. We first develop a token-level encoder to embed original features and mathematical operations, capturing token-level transformation patterns. Next, we aggregate token embeddings into concept embeddings according to the correspondence between original features, operations, and generated concepts. We then apply a concept-level encoder with self-attention pooling mechanism to eliminate order sensitivity, obtaining permutation-invariant global embeddings. To effectively train and optimize the encoders, we introduce a concept-level decoder and a token-level decoder to reconstruct concept embeddings and original feature-operation tokens, respectively. After obtaining the global embedding space, we employ a policy-guided multi-objective search strategy to explore this embedding space, guided by maximizing downstream task performance and minimizing the length of feature transformation sequences. The identified optimal embedding is then decoded into its corresponding feature transformation sequence using the trained decoder. Finally, this reconstructed sequence is applied to the original feature space, resulting in an optimized feature space with improved predictive performance.

## 3.2 Hierarchical Feature Transformation Knowledge Modeling

**Why Hierarchical Knowledge Modeling Matters.** Feature transformation inherently involves intricate hierarchical relationships among original features, mathematical operations, and higher-level generated concepts. However, existing generative intelligence-based methods typically treat the entire feature transformation sequence as an indistinguishable whole, neglecting these hierarchical distinctions. This oversimplification leads to inadequate modeling of transformation knowledge, reducing the discriminative power of the continuous embedding space, and ultimately resulting in suboptimal feature spaces. To address them, we propose a hierarchical modeling module in PHER, explicitly capturing both token-level relationships (individual feature-operation interactions) and concept-level relationships (aggregated higher-level concepts) within the learned embedding space.

**Reinforcement Transformation-Accuracy Training Data Collection.** To learn an effective embedding space of feature transformation, we collect $n$ feature transformation records using an RL-based method (Wang et al., 2022), denoted by $\{(\mathbf{\Gamma_i}, v_i)\}_{i=1}^n$, where $\mathbf{\Gamma_i}$ represents a feature transformation sequence (e.g., $log(f_1), -f_3, ...$) and $v_i$ denotes the corresponding model performance. In this method, three collaborative reinforced agents select two candidate features and one operation per iteration to generate new features. The entire procedure is optimized to maximize the downstream ML task performance. For more details, please refer to the referenced paper.

**Token Level Feature Transformation Embedding.** After collecting large-scale feature transformation sequence-accuracy pairs $\{(\mathbf{\Gamma_i}, v_i)\}_{i=1}^n$, we train the token-level encoder and decoder to capture token-level transformation patterns from feature-operation token sequences.

*Encoder $\phi_{tok}$:* The token encoder aims to learn a mapping function $\phi_{tok}$ that converts transformation sequence $\mathbf{\Gamma} \in \mathbb{R}^{1 \times N}$ to token embeddings $\mathbf{E}$, denoted by $\mathbf{E} = \phi_{tok}(\mathbf{\Gamma}) \in \mathbb{R}^{N \times d_{tok}}$, where $N$ is the length of the input transformation sequence $\mathbf{\Gamma}$ and $d_{tok}$ is the hidden size of token embedding. In PHER, we adopt two Transformer encoder layers (Vaswani et al., 2017) as the token encoder.

*Decoder $\psi_{tok}$:* The token decoder aims to reconstruct the transformation sequence $\mathbf{\Gamma}$ from the learned token embedding $\mathbf{E}$, denoted by $\mathbf{\Gamma} = \psi_{tok}(\mathbf{E})$. In PHER, we adopt two Transformer decoder layers (Vaswani et al., 2017) as the token decoder.

**Why Ensuring Permutation Invariance in Concept-Level Embeddings Matters.** While token-level encoders capture low-level interactions between features and operations, they struggle to model high-level transformation patterns. Feature transformations often yield feature abstractions (i.e. concepts) whose ordering does not affect the quality of the resulting feature space or the predictive performance of downstream tasks. However, existing methods encode these concepts in an order-sensitive manner, introducing permutation bias into the embedding space and limiting search effectiveness. To address this, we propose a permutation-invariant concept encoder-decoder with self-attention pooling to eliminate order sensitivity and enable more robust representation learning.

*Encoder $\phi_{con}$ :* The concept encoder $\phi_{con}$ aims to encode the generated concepts into permutation-invariant global embeddings. To ensure permutation invariance, we adopt a self-attention pooling mechanism (Lee et al., 2019) in $\phi_{con}$. This structure is inherently permutation-invariant, ensuring that any permutation of the generated concepts results in an identical global embedding. Specifically, we first adopt a mean pooling layer to aggregate the token embeddings $\mathbf{E}$ into concept embeddings $\mathbf{G}$ according to the relationship between concept and its affiliated feature indices and mathematical operations, serving as the aggregation layer in Figure 2. For instance, suppose concepts are derived from the transformation $(f_1 + f_7, log(f_2), ...)$, then corresponding concept embeddings are obtained by averaging the token embeddings of the involved features and mathematical operations, denoted by $\mathbf{G} = (\mathbf{G}_1, \mathbf{G}_2, ...)$, where $\mathbf{G}_1 = \text{Mean}(\mathbf{E}_{f_1}, \mathbf{E}_{plus}, \mathbf{E}_{f_7})$ and $\mathbf{G}_2 = \text{Mean}(\mathbf{E}_{f_2}, \mathbf{E}_{log})$. Then, we initialize a set of $k$ learnable seed vectors $\mathcal{S} \in \mathbb{R}^{k \times d_{seed}}$ using Xavier uniform initialization, where $k$ is the number of seed vectors and $d_{seed}$ is the hidden size of each seed vector. These learnable seed vectors serve as queries in the multi-head attention module, attending over the token embeddings $\mathbf{E}$ which serve as keys and values to extract high-level semantic concept representations. The output is then combined with the original seed vectors through a residual connection, followed by a linear layer. This yields the global embedding, denoted by: $\mathbf{G}' = \phi_{con}(\mathbf{G}) = \text{Linear}(\mathcal{S} + \text{Multihead}(\mathcal{S}, \mathbf{G}, \mathbf{G})) \in \mathbb{R}^{k \times d_{global}}$, where $d_{global}$ is the hidden size of global embedding.

*Decoder $\psi_{con}$ :* The concept decoder $\psi_{con}$ aims to reconstruct concept embedding $\mathbf{G}$ from the global embedding $\mathbf{G}'$, denoted by $\mathbf{G} = \psi_{con}(\mathbf{G}')$. In PHER, we adopt the same architecture for the concept decoder as used in the concept encoder, ensuring symmetric abstraction and reconstruction.

**Hierarchical Optimization Procedure:** To effectively train the hierarchical encoder-decoder models, we design a three-stage training strategy. Each stage progressively captures feature transformation knowledge, from token-level to concept-level.

*Stage 1: Token-Level Training.* In the first stage, we train the token encoder $\phi_{tok}$ and decoder $\psi_{tok}$ to reconstruct the input transformation sequence $\mathbf{\Gamma}$. Specifically, given a sequence $\mathbf{\Gamma}$, the token encoder first produces token embedding $\mathbf{E} = \phi_{tok}(\mathbf{\Gamma})$. The token decoder then reconstructs the original feature transformation sequence from the token embedding $\hat{\mathbf{\Gamma}} = \psi_{tok}(\mathbf{E})$. We optimize the token-level model by minimizing the negative log-likelihood loss: $\mathcal{L}_{tok} = -\log P_{\psi_{tok}}(\mathbf{\Gamma} \mid \mathbf{E})$.

*Stage 2: Concept-Level Training.* After training the token-level model, we freeze its parameters and train the concept encoder $\phi_{con}$ and decoder $\psi_{con}$ to learn high-level permutation-invariant global embeddings. Specifically, given a feature transformation sequence $\mathbf{\Gamma}$, we first utilize the well-trained token encoder to obtain token embedding $\mathbf{E}$. Then, we input $\mathbf{E}$ into the aggregation layer to generate the concept embedding $\mathbf{G}$ based on the relationships between concept and its affiliated feature indices and mathematical operations. Thereafter, $\mathbf{G}$ is input into $\phi_{con}$ to obtain the global embedding $\mathbf{G}' = \phi_{con}(\mathbf{G})$. Finally, the global embedding $\mathbf{G}'$ is decoded back to concept embedding $\hat{\mathbf{G}} = \psi_{con}(\mathbf{G}')$ We optimize the concept-level model by minimizing the Mean Squared Error (MSE) between the reconstructed $\hat{\mathbf{G}}$ and original concept embedding $\mathbf{G}$, denoted by: $\mathcal{L}_{con} = \text{MSE}(\mathbf{G}, \hat{\mathbf{G}}) = \text{MSE}(\mathbf{G}, \psi_{con}(\phi_{con}(\mathbf{G})))$.

*Stage 3: Token-Concept Alignment.* In the final stage, we freeze the concept-level model and tune the parameters of token encoder and decoder to improve the accuracy of end-to-end token reconstruction. Formally, given an transformation sequence $\mathbf{\Gamma}$, we obtain the token embedding $\mathbf{E} = \phi_{tok}(\mathbf{\Gamma})$. Then, $\mathbf{E}$ is input into the well-trained concept encoder $\phi_{con}^*$, the well-trained concept decoder $\psi_{con}^*$ and the token decoder in sequence to reconstruct the feature transformation sequence. We optimize the following loss during the alignment stage: $\mathcal{L}_{align} = -\log P_{\psi_{tok}}(\mathbf{\Gamma} \mid \psi_{con}^*(\phi_{con}^*(\phi_{tok}(\mathbf{\Gamma}))))$. This stage aligns token embeddings with the fixed concept-level abstractions, improving the coherence and semantic quality of the reconstructed transformation sequence.

## 3.3 POLICY-GUIDED MULTI-OBJECTIVE SEARCH

**Why selecting search seeds matters.** Once the global embedding space has converged, we employ a policy-guided multi-objective search strategy in the learned global embedding space to identify the optimal global embedding that achieves maximum downstream task performance and minimum transformation length. Inspired by the significance of initialization for deep neural networks, good starting points are crucial for accelerating the search process and enhancing its performance. Thus, we rank all collected records by model performance and select the top-K as search seeds.

**Policy-guided multi-objective search.** Existing methods assume the learned embedding space is convex, relying on gradient-ascent search strategy to identify the global optimal embedding. However, due to the complex interactions between features and mathematical operations, the embedding space is highly non-convex in practice, increasing the risk of becoming trapped in local optima and resulting suboptimal feature transformation sequences. Thus, to perform effective exploration and search process without strong convexity assumptions, we employ Proximal Policy Optimization (PPO) (Schulman et al., 2017) method, which is a widely used policy gradient algorithm for multi-objective optimization. PPO stabilizes policy updates by clipping and trust region mechanisms, effectively balancing exploration and exploitation in high-dimensional and non-convex embedding spaces. In PHER, the PPO agent learns to explore the continuous global embedding space by selecting promising global embeddings based on the downstream task performance and transformation sequence length. Specifically, given a global embedding $\mathbf{G}'$, the PPO agent $\mathcal{A}$ aims to manipulate it to generate the optimal global embedding $\mathbf{G}'_{opt}$, denoted by: $\mathcal{A}(\mathbf{G}') = \mathbf{G}'_{opt}$.

**Reward $\mathcal{R}$:** To balance the performance and the total length of the transformed sequence, we propose a weighted reward function, denoted by: $\mathcal{R} = \lambda(\mathcal{M}(X[\mathbf{\Gamma}^+]) - \mathcal{M}(X[\mathbf{\Gamma}])) + (1-\lambda)\mathcal{N}[\mathbf{\Gamma}^+]$, where $\mathcal{M}$ is the downstream ML task, $\mathbf{\Gamma}$ is the original transformation sequence, $\mathbf{\Gamma}^+$ is the searched transformation sequence, $\mathcal{N}[\mathbf{\Gamma}^+]$ is a normalized length penalty of $\mathbf{\Gamma}^+$, and $\lambda$ is the trade-off hyperparameter to balance the performance improvement and the total length of the transformed sequence.

**Solving the Optimization Problem.** We adopt a multi-objective optimization strategy to train the PPO agent. Specifically, the policy is updated by maximizing a clipped surrogate objective, which approximates the cumulative discounted reward obtained throughout the iterative search process, while constraining the step size of each update. To enable the PPO agent to identify more informative embeddings with higher model performance and fewer transformation opera-

tions, we adopt an actor-critic model architecture. We optimize the critic network by minimizing the difference between the observed cumulative discounted rewards and their predictions, denoted by: $\mathcal{L}_{critic} = \frac{1}{T}\sum_{t=1}^{T}(V(\mathbf{s}_t) - G_t)^2$, where $T$ is the trajectory length, $\mathbf{s}_t$ is the state at time step $t$, $V(\mathbf{s}_t)$ is the predicted value from the critic, $G_t$ is the cumulative discounted return, and $\gamma \in [0 \sim 1]$ is the discounted factor. The actor network is optimized by maximizing a clipped surrogate objective while constraining the step size of each policy update, denoted by: $\mathcal{L}_{actor} = \hat{\mathbb{E}}_t\left[min(r_t(\theta)\hat{A}_t, clip(r_t(\theta), 1-\epsilon, 1+\epsilon)\hat{A}_t)\right]$, where $\epsilon$ is the clipping ratio hyperparameter, $r_t(\theta)$ is the probability ratio, and $\hat{A}_t$ is the estimated advantage. Once the actor network converges, the resulting policy $\pi^*$ guides the global embedding $\mathbf{G}'$ in the learned global embedding space $\mathbb{G}$ toward regions that yield higher downstream task performance but fewer transformation operations, without relying on any convexity assumptions. Then, we reconstruct the transformation sequence $\mathbf{\Gamma}^+$ from the enhanced global embedding $\mathbf{G}'_+$ using the well-trained concept decoder and token decoder. Next, we obtain enhanced feature spaces using the enhanced transformation sequences $[\mathbf{\Gamma}_1^+, \mathbf{\Gamma}_2^+, ..., \mathbf{\Gamma}_K^+]$ and input them into the downstream ML task to evaluate their performance. Finally, the feature transformation sequence achieving the highest performance is selected as the optimal sequence, denoted as $\mathbf{\Gamma}^*$. This sequence serves as the guiding blueprint for refining the original feature space, thus producing an enhanced feature space.

## 4 EXPERIMENTS

To enhance the reproducibility of PHER, we provide hyperparameter settings and experimental platform information in Appendix C.1 and C.2.

### 4.1 DATASETS AND EVALUATION METRICS

We evaluate PHER on 19 publicly accessible datasets from UCI Public (2022b), LibSVM Chih-Jen (2022), Kaggle Howard (2022), and OpenML Public (2022a), including 14 classification tasks and 5 regression tasks. Appendix B, Table 3 summarizes the detailed statistics of these dataset. To ensure consistent and stable results, we adopt Random Forest as the unified downstream model and evaluate performance using five-fold cross-validation. Each experiment is independently repeated five times with different random seeds, and we report the mean and standard deviation. All experiments follow a hold-out evaluation protocol. For regression tasks, we report 1-Relative Absolute Error (1-RAE), 1-Mean Absolute Error (1-MAE), 1-Mean Squared Error (1-MSE), and 1-Root Mean Squared Error (1-RMSE). For classification tasks, we use F1-score, Precision, Recall, and ROC/AUC.

### 4.2 BASELINE MODELS

We compare our method with nine widely-used feature transformation methods: (1) RFG generates new feature-operation-feature transformation records by randomly selecting candidate features and operations. (2) ERG first applies operations to each feature to expand the feature space and then selects informative features as new features. (3) LDA Blei et al. (2003) is a matrix factorization-based dimensionality reduction method that projects data into a low-dimensional space. (4) AFAT Horn et al. (2019a) iteratively applies transformations and performs multi-step feature selection to identify informative ones. (5) NFS Chen et al. (2019b) uses a recurrent neural network-based controller trained with reinforcement learning to sequentially model and optimize the transformation process for each feature. (6) TTG Khurana et al. (2018b) formulates feature transformation as a graph exploration problem and employs reinforcement learning to discover optimal transformation paths within the graph. (7) GRFG Wang et al. (2022) employs three reinforced agents with a feature grouping strategy to perform feature generation in a cascaded manner. (8) DIFER Zhu et al. (2022b) encodes randomly generated transformation sequences into a continuous embedding space and employs greedy gradient-based search to identify the best transformed features. (9) MOAT Wang et al. (2023) embeds RL-collected transformation sequences into continuous embeddings using postfix expression and conduct gradient-ascent search with the beam search strategy. Besides, we developed three variants of PHER to evaluate the impact of each technical component: (i) **PHER**$^{-c}$ removes the concept encoder and decoder. (ii) **PHER**$^{-p}$ replaces the permutation-invariant concept encoder and decoder with permutation-sensitive self-attention layers. (iii) **PHER**$^{-g}$ replaces the policy-guided search with Genetic Algorithm (GA). To ensure fair evaluation, we randomly divide each dataset into 80% for training and 20% for testing. This experimental setting prevents any test data leakage and ensures a more reliable comparison of feature transformation performance.

Table 1: Overall performance comparison. In this table, the best and second-best results are highlighted in red and blue, respectively.(**Higher values indicate better performance.**)

| Dataset | C/R | Original | RDG | ERG | LDA | AFAT | NFS | TTG | GRFG | DIFER | MOAT | PHER |
|---|---|---|---|---|---|---|---|---|---|---|---|---|
| Higgs Boson | C | 0.696 | $0.700^{\pm0.001}$ | $0.704^{\pm0.003}$ | $0.511^{\pm0.011}$ | $0.695^{\pm0.001}$ | $0.697^{\pm0.002}$ | $0.699^{\pm0.003}$ | $0.701^{\pm0.002}$ | $0.669^{\pm0.001}$ | $0.699^{\pm0.002}$ | $0.706^{\pm0.002}$ |
| Amazon Employee | C | 0.930 | $0.931^{\pm0.001}$ | $0.935^{\pm0.001}$ | $0.914^{\pm0.002}$ | $0.933^{\pm0.002}$ | $0.932^{\pm0.001}$ | $0.930^{\pm0.001}$ | $0.932^{\pm0.001}$ | $0.929^{\pm0.001}$ | $0.933^{\pm0.002}$ | $0.936^{\pm0.002}$ |
| PimaIndian | C | 0.776 | $0.760^{\pm0.007}$ | $0.762^{\pm0.004}$ | $0.729^{\pm0.061}$ | $0.760^{\pm0.011}$ | $0.759^{\pm0.014}$ | $0.750^{\pm0.019}$ | $0.754^{\pm0.011}$ | $0.760^{\pm0.013}$ | $0.807^{\pm0.011}$ | $0.828^{\pm0.003}$ |
| SpectF | C | 0.760 | $0.760^{\pm0.001}$ | $0.759^{\pm0.018}$ | $0.665^{\pm0.119}$ | $0.782^{\pm0.079}$ | $0.760^{\pm0.001}$ | $0.775^{\pm0.014}$ | $0.818^{\pm0.001}$ | $0.766^{\pm0.002}$ | $0.912^{\pm0.014}$ | $0.929^{\pm0.003}$ |
| SVMGuide3 | C | 0.778 | $0.791^{\pm0.008}$ | $0.817^{\pm0.015}$ | $0.635^{\pm0.049}$ | $0.789^{\pm0.009}$ | $0.786^{\pm0.004}$ | $0.791^{\pm0.014}$ | $0.812^{\pm0.025}$ | $0.773^{\pm0.021}$ | $0.845^{\pm0.001}$ | $0.848^{\pm0.003}$ |
| German Credit | C | 0.649 | $0.667^{\pm0.008}$ | $0.684^{\pm0.011}$ | $0.597^{\pm0.058}$ | $0.640^{\pm0.032}$ | $0.663^{\pm0.017}$ | $0.663^{\pm0.019}$ | $0.683^{\pm0.013}$ | $0.656^{\pm0.018}$ | $0.729^{\pm0.001}$ | $0.736^{\pm0.007}$ |
| Credit Default | C | 0.802 | $0.805^{\pm0.001}$ | $0.804^{\pm0.002}$ | $0.743^{\pm0.009}$ | $0.804^{\pm0.001}$ | $0.803^{\pm0.002}$ | $0.803^{\pm0.002}$ | $0.806^{\pm0.003}$ | $0.796^{\pm0.005}$ | $0.808^{\pm0.001}$ | $0.810^{\pm0.001}$ |
| Messidor_features | C | 0.633 | $0.685^{\pm0.053}$ | $0.665^{\pm0.025}$ | $0.463^{\pm0.084}$ | $0.656^{\pm0.005}$ | $0.657^{\pm0.011}$ | $0.662^{\pm0.009}$ | $0.692^{\pm0.033}$ | $0.660^{\pm0.007}$ | $0.694^{\pm0.005}$ | $0.744^{\pm0.003}$ |
| Wine Quality Red | C | 0.456 | $0.514^{\pm0.016}$ | $0.494^{\pm0.005}$ | $0.401^{\pm0.061}$ | $0.480^{\pm0.042}$ | $0.467^{\pm0.021}$ | $0.464^{\pm0.010}$ | $0.470^{\pm0.009}$ | $0.476^{\pm0.018}$ | $0.559^{\pm0.002}$ | $0.562^{\pm0.003}$ |
| Wine Quality White | C | 0.536 | $0.524^{\pm0.001}$ | $0.524^{\pm0.001}$ | $0.437^{\pm0.015}$ | $0.516^{\pm0.032}$ | $0.533^{\pm0.010}$ | $0.529^{\pm0.002}$ | $0.534^{\pm0.019}$ | $0.507^{\pm0.025}$ | $0.536^{\pm0.010}$ | $0.553^{\pm0.002}$ |
| SpamBase | C | 0.927 | $0.924^{\pm0.001}$ | $0.920^{\pm0.001}$ | $0.885^{\pm0.030}$ | $0.917^{\pm0.001}$ | $0.922^{\pm0.003}$ | $0.922^{\pm0.003}$ | $0.922^{\pm0.005}$ | $0.912^{\pm0.016}$ | $0.932^{\pm0.001}$ | $0.936^{\pm0.002}$ |
| AP-omentum-ovary | C | 0.845 | $0.845^{\pm0.001}$ | $0.814^{\pm0.001}$ | $0.710^{\pm0.134}$ | $0.845^{\pm0.125}$ | $0.845^{\pm0.001}$ | $0.845^{\pm0.001}$ | $0.849^{\pm0.001}$ | $0.833^{\pm0.031}$ | $0.845^{\pm0.002}$ | $0.883^{\pm0.001}$ |
| Lymphography | C | 0.141 | $0.108^{\pm0.001}$ | $0.129^{\pm0.009}$ | $0.144^{\pm0.132}$ | $0.150^{\pm0.133}$ | $0.170^{\pm0.035}$ | $0.174^{\pm0.040}$ | $0.182^{\pm0.013}$ | $0.150^{\pm0.116}$ | $0.267^{\pm0.035}$ | $0.410^{\pm0.105}$ |
| MNIST fashion | C | 0.712 | $0.714^{\pm0.001}$ | $0.716^{\pm0.001}$ | $0.510^{\pm0.020}$ | $0.699^{\pm0.022}$ | $0.710^{\pm0.002}$ | $0.711^{\pm0.004}$ | $0.728^{\pm0.001}$ | $0.717^{\pm0.002}$ | $0.729^{\pm0.001}$ | $0.735^{\pm0.001}$ |
| Housing Boston | R | 0.415 | $0.420^{\pm0.031}$ | $0.418^{\pm0.001}$ | $0.021^{\pm0.002}$ | $0.423^{\pm0.005}$ | $0.424^{\pm0.002}$ | $0.421^{\pm0.011}$ | $0.404^{\pm0.007}$ | $0.381^{\pm0.017}$ | $0.465^{\pm0.006}$ | $0.538^{\pm0.005}$ |
| Airfoil | R | 0.519 | $0.520^{\pm0.001}$ | $0.519^{\pm0.001}$ | $0.207^{\pm0.065}$ | $0.509^{\pm0.001}$ | $0.519^{\pm0.002}$ | $0.521^{\pm0.001}$ | $0.521^{\pm0.003}$ | $0.558^{\pm0.002}$ | $0.627^{\pm0.011}$ | $0.645^{\pm0.001}$ |
| Openml_589 | R | 0.510 | $0.548^{\pm0.032}$ | $0.610^{\pm0.001}$ | $0.034^{\pm0.079}$ | $0.509^{\pm0.002}$ | $0.506^{\pm0.004}$ | $0.502^{\pm0.002}$ | $0.627^{\pm0.006}$ | $0.463^{\pm0.005}$ | $0.656^{\pm0.001}$ | $0.660^{\pm0.006}$ |
| Openml_618 | R | 0.469 | $0.444^{\pm0.026}$ | $0.543^{\pm0.039}$ | $0.030^{\pm0.076}$ | $0.473^{\pm0.002}$ | $0.471^{\pm0.004}$ | $0.472^{\pm0.001}$ | $0.562^{\pm0.101}$ | $0.408^{\pm0.036}$ | $0.692^{\pm0.002}$ | $0.693^{\pm0.001}$ |
| Openml_620 | R | 0.510 | $0.494^{\pm0.019}$ | $0.541^{\pm0.008}$ | $0.026^{\pm0.013}$ | $0.520^{\pm0.001}$ | $0.509^{\pm0.005}$ | $0.513^{\pm0.004}$ | $0.568^{\pm0.013}$ | $0.442^{\pm0.004}$ | $0.643^{\pm0.003}$ | $0.652^{\pm0.003}$ |

\* We evaluated classification (C) and regression (R) tasks in terms of F1-Score and 1-RAE, respectively.

\* The standard deviation is computed based on the results of 5 independent runs.

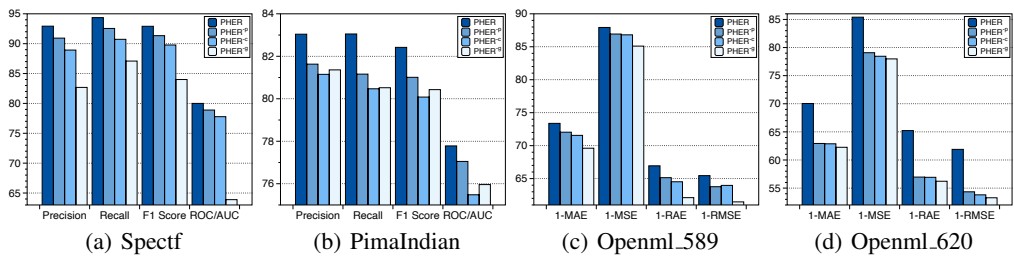

| (a) Spectf | (b) PimaIndian | (c) Openml_589 | (d) Openml_620 |
|---|---|---|---|

Figure 3: The influence of hierarchical modeling (PHER$^{-c}$), permutation invariance (PHER$^{-p}$), and policy-guided search (PHER$^{-g}$) in PHER.

## 4.3 PERFORMANCE EVALUATION

**Overall Performance.** Table 1 reports the performance comparison between PHER and baseline models across 19 datasets, using F1-score for classification tasks and 1-RAE for regression tasks. All experimental results are obtained by independently running five times with different random seeds. We observed that PHER outperforms the other baseline models on all datasets. There are three potential reasons for this observation: 1) The hierarchical modeling architecture captures both low-level and high-level relationships between features, encoding diverse feature transformation knowledge and building a more informative global embedding space; 2) The permutation-invariant concept encoder-decoder framework eliminate order sensitivity among concepts, constructing a global unbiased embedding space; 3) The policy-based RL agent effectively explores the global embedding space and identifies superior feature transformation embeddings, overcoming non-convex challenges and converging to the global optimal embedding point. Thus, this experiment demonstrates the effectiveness of PHER in transforming feature spaces across various types of tasks.

**Ablation Study.** We conduct this experiment to study the impact of the hierarchical modeling module, permutation-invariant embedding, and policy-guided search on the model performance. We developed three model variants of PHER: 1) PHER$^{-c}$ removes the concept encoder-decoder model; 2) PHER$^{-p}$ replaces the permutation-invariant concept encoder and decoder with permutation-sensitive self-attention layers. 3) PHER$^{-g}$ replaces the policy-guided search with Genetic Algorithm (GA). We randomly select two classification tasks (SpectF and PimaIndian) and two regression tasks (Openml_589 and Openml_620) to show the comparison results. We evaluate the model performance on each type of task from four different perspectives. For classification tasks, we use F1-score, Precision, Recall, and ROC/AUC. For regression tasks, we use 1-RAE, 1-MAE, 1-MSE, and 1-RMSE. As shown in figure 3, the performance of PHER is consistently higher than PHER$^{-c}$, PHER$^{-p}$, and PHER$^{-g}$. The potential reasons for this observation are: 1) the hierarchical modeling module captures both low-level feature relationships and high-level concepts, preserving more informative feature transformation knowledge within the global embedding space; 2) the permutation-invariant mechanism removes permutation noise among generated concepts in the embedding space, facilitating more effective exploration by the RL agent. 3) the RL agent conducts effective exploration in the embedding space, eliminating the reliance on convexity assumptions and avoiding convergence to local optima. Thus, this experiment evaluates the significance of the hierarchical modeling module, permutation-invariant embedding, and policy-guided search.

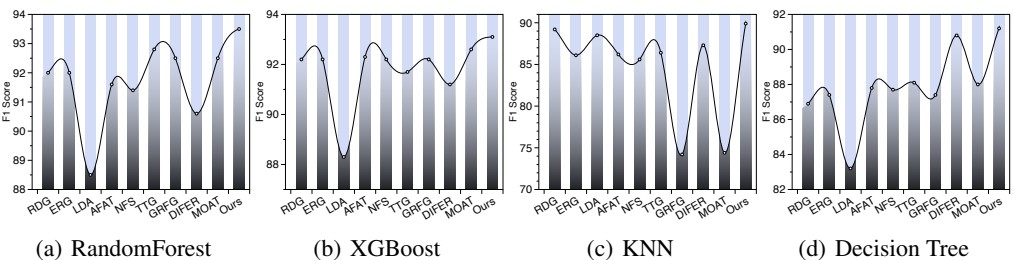

Figure 4: Robustness check of PHER with distinct ML models on SpamBase in terms of F1-score.

**Robustness Check.** We evaluate PHER using various downstream ML models, including Random Forest (RF), XGBoost (XGB), K-Nearest Neighborhood (KNN), and Decision Tree (DT), to assess the robustness of PHER. Figure 4 shows the comparison results on SpamBase in terms of the F1-score. We observed that PHER consistently outperforms all baseline methods across different downstream ML models. A potential reason is that the RL-based data collector explores diverse feature transformation records guided by validation performance rather than any specific classifier These records further enable the hierarchical model to encode task-relevant knowledge and characteristics into the global embedding space. Finally, the policy-guided agent leverages these information to effectively search the embedding space. The results on Support Vector Machine (SVM), Ridge, and LASSO are reported in Appendix D.1, Figure 8.

**Search Seeds.** To observe the impacts of search seeds on the RL-based search process, we replace the top K historical feature transformation records with K random records. We conduct this experiment on eight randomly chosen datasets and report the average F1-score over five independent runs. As shown in Figure 5, search initialized with high-quality seeds consistently outperforms search with random initialization. A potential reason for this observation is that the RL agent can leverage informative seeds to anchor the search within promising regions of the embedding space, effectively narrowing the search scope and accelerating convergence. Thus, this experiment shows the necessity of the search seeds in PHER.

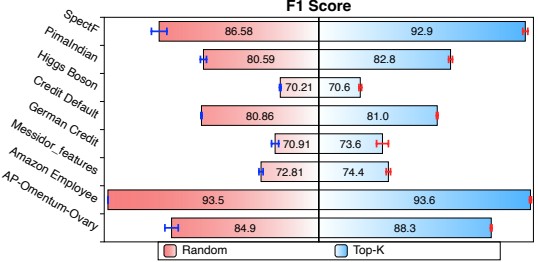

Figure 5: The influence of search seeds.

**Model Scalability.** To analysis model scalability, we compare the length of feature transformation sequence and downstream task performance of PHER and the SOTA model MOAT. Figure 6 (a) shows that PHER produces significantly shorter feature transformation sequences, showing its ability to discover compact and effective feature transformation sequences. From Figure 6 (b), we found that PHER consistently outperforms MOAT across all datasets. These observations suggest that the RL agent in PHER effectively optimizes the learned embeddings by jointly maximizing task performance and minimizing transformation sequence length. Thus, this experiment demonstrates that PHER consistently achieves strong performance and compactness across diverse datasets, highlighting the scalability of PHER.

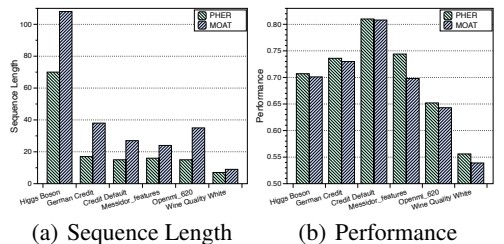

(a) Sequence Length    (b) Performance

Figure 6: Model scalability comparison between our method and the SOTA model MOAT.

**Hyperparameter Sensitivity Analysis.** We assess the hyperparameter sensitivity of PHER on the SpectF dataset by varying the clipping ratio $\epsilon$ and the reward trade-off $\lambda$ from 0.1 to 0.9, where $\epsilon$ stabilizes training by controlling the magnitude of policy updates and $\lambda$ balances task performance against transformation sequence length. Figure 7, demonstrates the overall experimental results in terms of Precision, Recall and F1-Score. We found that PHER achieves optimal performance when $\epsilon = 0.2$ and $\lambda = 0.9$. This aligns with prior PPO studies, where $\epsilon = 0.2$ is a setting widely adopted in standard reinforcement learning benchmarks Schulman et al. (2017).

Furthermore, we found that the model performance remains relatively stable across a broad range of $\lambda$, with slight improvements as $\lambda$ approaches 0.9. Thus, this experiment demonstrates the model sensitivity to hyperparameter settings and provides practical guidance on hyperparameter configuration. To provide a comprehensive evaluation of PHER, we analyze the time complexity (Appendix D.2), conduct a traceability case study (Appendix D.3), and include a study on the permutation sensitivity of the learned global embeddings (Section D.4).

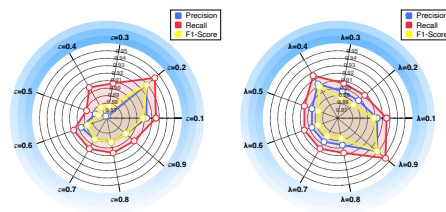

(a) Clipping Trade-off   (b) Reward Trade-off

Figure 7: Hyperparameter sensitivity.

## 5 RELATED WORKS

**Automated Feature Transformation (AFT)** can enhance the tabular feature spaces by applying mathematical operations to original features Chen et al. (2021); Kusiak (2001). Existing methods can be grouped into three main types: 1) expansion-reduction based approaches Kanter & Veeramachaneni (2015); Khurana et al. (2016b); Lam et al. (2017); Horn et al. (2019b); Khurana et al. (2016a) first expand the feature space through mathematical transformations and then reduce dimensionality by selecting informative features. However, these methods struggle to effectively capture complex feature compositions, resulting in suboptimal results. 2) evolution-evaluation approaches Wang et al. (2022); Khurana et al. (2018b); Tran et al. (2016); Zhu et al. (2022a); Zhang et al. (2022); Katz et al. (2016) combine feature generation and selection into a closed-loop learning system, using evolutionary algorithms or reinforcement learning (RL) to iteratively generate transformed features and retain the most effective ones. However, these methods incur high computational cost and unstable performance due to discrete decision-making. 3) Auto ML-based approaches Elsken et al. (2019); Li et al. (2021); He et al. (2021); Karmaker et al. (2021); Zhang et al. (2021); Wever et al. (2021); Bahri et al. (2022); Wang et al. (2021); Dor & Reich (2012); Egozi et al. (2008); Ren et al. (2023) formulate AFT as an AutoML problem, searching for transformation strategies alongside model optimization. For instance, Wang et al. (2023) embeds RL-collected transformation sequences into continuous embeddings using postfix expression and conducts gradient-ascent beam search to identify informative feature transformation embeddings.

**Comparison with Prior Literature.** While existing approaches have made substantial progress, current formulations generally operate at the feature level without capturing the inherent hierarchical structure in transformation processes. Moreover, the resulting embedding space is typically order-sensitive and optimized through gradient-based search method. In contrast, PHER incorporates hierarchical modeling to jointly preserve token-level interactions and concept-level abstractions and further employs a permutation-invariant mechanism to remove order-induced biases. Thereafter, we adopt a multi-objective policy-guided search strategy to explore the non-convex embedding space.

## 6 CONCLUSION REMARKS

In this paper, we propose a hierarchical feature transformation framework PHER that integrates permutation-invariant hierarchical modeling and multi-objective policy-guided search. In detail, we first develop a permutation-invariant hierarchical modeling module, including token-level and concept-level encoder-decoder models, to preserve the feature-operation token level and the generated-concept level feature transformation knowledge into a global embedding space. Within this module, we develop a self-attention pooling mechanism that symmetrically computes attention scores across all generated concepts to ensure permutation invariance. Then, we employ a multi-objective search strategy to explore the learned embedding space, overcoming the reliance on convex assumptions and mitigating the risk of being trapped in local optima. Finally, extensive experiments demonstrate several key insights: 1) hierarchical modeling structure significantly captures meaningful hierarchical interactions, enhancing the expressivity of the learned embedding space. 2) permutation-invariant module effectively mitigates order sensitivity, stabilizing the embedding space learning and search processes. 3) policy-guided RL search enables effective exploration, avoiding convergence to local optima and improving search robustness. These findings highlight the importance of permutation-invariant hierarchical modeling and robust exploration strategies for advancing automated feature transformation. In the future, a promising direction is to improve the computational efficiency and scalability of PHER, possibly by integrating lightweight concept modeling framework or refining the RL-based search strategy in the learned embedding space.

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

# Appendix

## A  SUMMARY OF NOTATIONS

To ensure clarity, we provide a summary of the mathematical notations used throughout the paper. Table 2 lists the key symbols along with their descriptions.

Table 2: Summary of Notations

| Notation | Description |
|---|---|
| $D = \{X, y\}$ | Dataset |
| $X$ | Input features |
| $y$ | Target variable |
| $\mathcal{O}$ | Operation set |
| $\mathcal{M}$ | Downstream task |
| $\Gamma_i$ | Feature transformation record $i$ |
| $v_i$ | Downstream task performance of feature transformation record $i$ |
| $\Gamma^+$ | Searched feature transformation record |
| $\Gamma^*$ | Optimal Feature transformation record |
| $\phi_{con}$ | Concept encoder |
| $\psi_{con}$ | Concept decoder |
| $\phi_{tok}^*$ | Well-trained token encoder |
| $\phi_{con}^*$ | Well-trained concept encoder |
| $\psi_{tok}^*$ | Well-trained token decoder |
| $\psi_{con}^*$ | Well-trained concept decoder |
| $\mathbf{E}$ | Token embedding |
| $\mathbf{G}$ | Concept embedding |
| $\hat{\mathbf{G}}$ | Reconstructed concept embedding |
| $\mathbf{G}'$ | Global embedding |
| $\mathbf{G}'_+$ | Enhanced global embedding |
| $\mathbf{G}'_{opt}$ | Optimal global embedding |
| $\mathbb{G}$ | Global embedding space |
| $\mathcal{S}$ | Learnable seed vector |
| $k$ | Number of seed vector |
| $N$ | Length of feature transformation record |
| $d_{seed}$ | Hidden size of seed vector |
| $d_{tok}$ | Hidden size of token embedding |
| $d_{global}$ | Hidden size of global embedding |
| $\mathcal{A}$ | PPO agent |
| $\mathbf{a}$ | Agent action |
| $\mathbf{s}$ | Agent state |
| $\mathcal{R}$ | Agent reward |
| $\lambda$ | Reward trade-off hyperparameter |
| $\epsilon$ | Clipping ratio hyperparameter |
| $T$ | Trajectory length |
| $V(\mathbf{s}_t)$ | Agent predicted reward |
| $G_t$ | Discounted reward |
| $\gamma$ | Discount vector |
| $\pi$ | Agent policy |
| $\pi^*$ | Optimal agent policy |
| $r_t(\theta)$ | Probability ratio |
| $\mathcal{L}_{tok}$ | Loss function of training token-level model |
| $\mathcal{L}_{con}$ | Loss function of training concept-level model |
| $\mathcal{L}_{align}$ | Loss function of aligning token-concept models |

## B  DATASET STATISTICS

Table 3 summarizes the details of the datasets used in our experiments. For each dataset, we report its source, number of samples, and the number of features. These datasets cover a diverse range

Table 3: Dataset Statistics Overview

| Dataset | Source | Samples | Features |
|---|---|---|---|
| Higgs Boson | UCIrvine | 50000 | 28 |
| Amazon Employee | Kaggle | 32769 | 9 |
| PimaIndian | UCIrvine | 768 | 8 |
| SpectF | UCIrvine | 267 | 44 |
| SVMGuide3 | LibSVM | 1243 | 21 |
| German Credit | UCIrvine | 1001 | 24 |
| Credit Default | UCIrvine | 30000 | 25 |
| Messidor_features | UCIrvine | 1150 | 19 |
| Wine Quality Red | UCIrvine | 999 | 12 |
| Wine Quality White | UCIrvine | 4900 | 12 |
| SpamBase | UCIrvine | 4601 | 57 |
| AP-omentum-ovary | OpenML | 275 | 10936 |
| Lymphography | UCIrvine | 148 | 18 |
| MNIST fashion | Kaggle | 10000 | 784 |
| Housing Boston | UCIrvine | 506 | 13 |
| Airfoil | UCIrvine | 1503 | 5 |
| Openml_589 | OpenML | 1000 | 25 |
| Openml_618 | OpenML | 1000 | 50 |
| Openml_620 | OpenML | 1000 | 25 |

of domains and scales, including both classification and regression tasks. Such diversity ensures a comprehensive evaluation of the proposed method across different scenarios.

## C  HYPERPARAMETER AND EXPERIMENTAL SETTINGS

### C.1  HYPERPARAMETER SETTINGS AND REPRODUCIBILITY

The operation set incorporates a diverse set of unary and binary transformations, including *square root*, *square*, *cosine*, *sine*, *tangent*, *exp*, *cube*, *log*, *reciprocal*, *quantile transformer*, *min-max scale*, *sigmoid*, *plus*, *subtract*, *multiply*, *divide*. We ran the RL-based data collector for 512 epochs to collect a substantial set of feature transformation–accuracy pairs. To enhance the diversity of training data, we randomly shuffled each transformation sequence 10 times. The hidden size of token embedding, concept embedding and global embedding are set as 128. To train the permutation-invariant hierarchical modeling module, we set batch size as 256, the step size as 0.001, the dimension of seed vectors as 128, respectively. To conduct effective search process, we used the top 20 feature transformation-accuracy records as starting points to search for the optimal transformation embeddings. To stabilize the search process, we set search epoch as 10, learning rate of the actor as 0.0003, learning rate of the critic as 0.001, reward trade-off as 0.9, reward discounted factor as 0.99, search step as 1000, and the clipping ratio as 0.2, respectively.

### C.2  EXPERIMENTAL PLATFORM INFORMATION

All experiments were conducted on the Windows 11 operating system, AMD Ryzen 5 5600X CPU, and NVIDIA GeForce RTX 3070Ti GPU, with the framework of Python 3.10.15 and PyTorch 2.5.1.

## D  EXPERIMENTAL RESULTS

### D.1  ROBUSTNESS CHECK

To complement the robustness analysis in Section 4.3, we report additional results on Support Vector Machine (SVM), Ridge Regression (Ridge), and LASSO models. Figure 8 summarizes the F1-scores obtained on the SpamBase dataset using these additional downstream ML models. The results further confirm the consistent superiority of PHER across different learning algorithms. Thus, this experiment demonstrates the robustness of PHER.

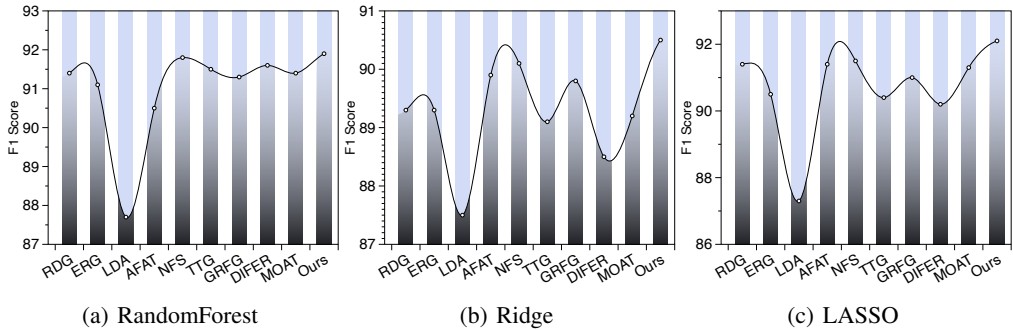

(a) RandomForest          (b) Ridge          (c) LASSO

Figure 8: Robustness check of PHER with distinct ML models on SpamBase dataset in terms of F1-score.

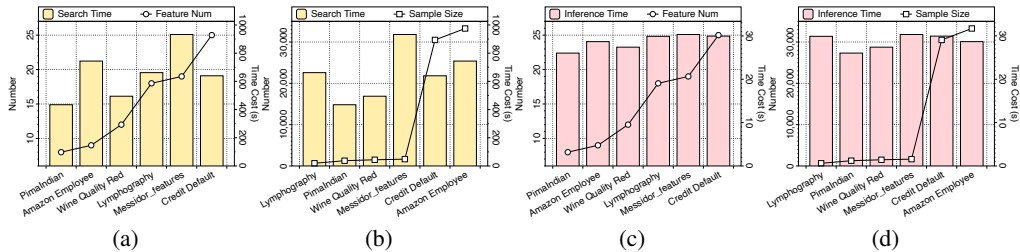

(a)        (b)        (c)        (d)

Figure 9: Time complexity of PHER in search time and inference time based on feature number and sample size.

## D.2 TIME COMPLEXITY

Reinforcement learning (RL) methods are often associated with high computational costs due to iterative exploration and delayed reward feedback. To evaluate the practical efficiency of PHER, we report the trends of the search time and inference time with respect to sample size and feature dimensionality on multiple datasets. Figure 9 shows the experimental results in terms of second (s). As shown in Figure 9 (a), search time increases moderately as the number of feature columns grows, which is expected since higher-dimensional feature transformations require additional encoding and evaluation steps. In contrast, Figure 9 (b) shows that search time remains largely stable as the sample size increases. The reason is that the PHER search procedure relies on the predicted performance from the downstream model, rather than repeatedly fitting the model from scratch on the entire dataset. Therefore, sample size has minimal influence on search runtime. Moreover, Figure 9 (c) and (d) demonstrate that the inference time of PHER remains relatively stable as both the feature dimensionality and sample size increase. This observation is consistent with our expectation, as we map the transformation sequence of varying lengths into a continuous space with uniform dimensionality, resulting in the stability of inference time of PHER. Therefore, this experiment demonstrates that PHER maintains stable and scalable time complexity, despite its hierarchical modeling and RL-based search.

## D.3 TRACEABILITY CASE STUDY

We conduct this experiment to evaluate the traceability of PHER. We rank the top 10 significant features for prediction in both the original feature set and PHER generated feature set of the Wine Quality Red dataset. Figure 10 visualize the experiment results, where larger bar indicates higher feature importance. We observed that approximately 70% crucial features in the new feature set are generated by PHER. The new generated feature space enhances the downstream ML performance by 23.7%. There are two potential reasons for this observation: 1) the hierarchical modeling module in PHER effectively captures the inherent hierarchical relationships from low-level features and operations to high-level concepts. 2) the policy-guided multi-objective search strategy effectively explores the learned global embedding space and identifies the superior feature transformation sequence, overcoming the non-convex challenge and converging to the global optimal embedding point. Furthermore, we found that '[alcohol]' is the most important feature in the original set. This

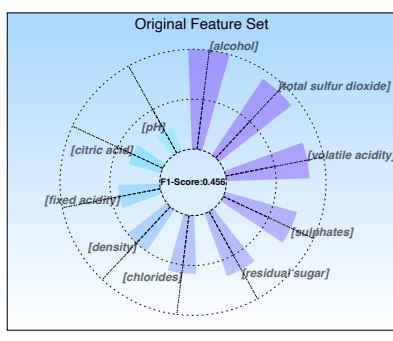 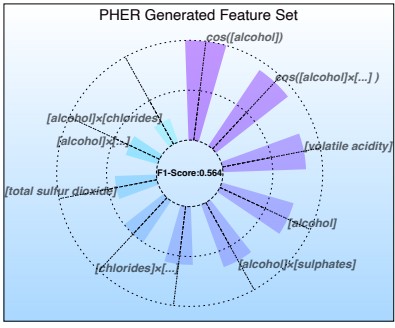

(a) Original Feature Space    (b) PHER Generated Feature Space

Figure 10: Comparison of traceability on the original feature set and selected feature subset.

aligns with domain knowledge, as alcohol is known to be one of the most influential factors in determining red wine quality. PHER not only identifies this essential feature but also generates a variety of composited features based on '[alcohol]', which further enhance the predictive performance. This observation demonstrates that PHER is capable of identifying the importance of individual features while also deriving new informative representations that align with domain semantics and enhance downstream performance. Such new informative features empower domain experts to trace the origins of transformed features and derive novel analytical rules for assessing red wine quality. Thus, this case study demonstrates the traceability and interpretability of PHER.

### D.4 PERMUTATION SENSITIVITY

We conduct this experiment to study the permutation sensitivity of learned global embeddings. We randomly select four datasets and visualize the global embeddings of the original concept embeddings and their corresponding permuted versions. For each dataset, we randomly choose five samples as representatives and further permute the concept embedding of each sample 20 times. Specifically, for each example, we first obtain its concept embedding set and create multiple permuted variants by randomly shuffling the order of the concept embeddings. Both the original and permuted concept embedding sets are then passed through the trained concept encoder to obtain their global embeddings. Thereafter, we visualize these embeddings using t-SNE. Figure 11 shows the visualization results, where each color represents the global embedding of one concept embedding set together with the embeddings derived from its permuted variants. We find that the global embeddings derived from permuted concept orders consistently cluster around the original embedding. A potential reason for this observation is that the learned global embedding space inherently captures permutation-invariant structure. Therefore, this experiment confirms that our proposed hierarchical encoder-decoder effectively removes permutation bias.

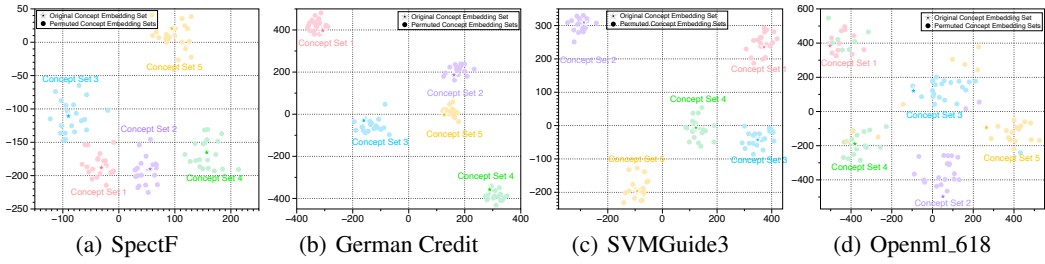

(a) SpectF    (b) German Credit    (c) SVMGuide3    (d) Openml_618

Figure 11: Visualization of the global embeddings obtained from each original concept embedding set and its randomly permuted variants.

# E LIMITATIONS AND FUTURE WORK

Extensive experimental results demonstrate that PHER delivers substantial performance improvements across diverse datasets and exhibits strong generalizability across heterogeneous tasks. Future work could explore several promising directions to further enhance the capability and adaptability of PHER across broader scenarios: 1) PHER incorporates a hierarchical modeling module to explicitly capture both token-level relationships and concept-level relationships. One potential direction for improvement might be to adopt a lightweight alternative to further enhance the efficiency of PHER when scaling to large-scale datasets. 2) PHER employs a policy-guided multi-objective search strategy to explore the learned embedding space and identify the optimal feature transformation sequence. Another potential direction for improvement might be to simplify the trajectory collection process or adopt a more compact representation of the search space to further reduce computational overhead. These directions offer opportunities to further enhance the capability and adaptability of PHER in more complex real-world scenarios, which are also our future research directions.

