# OpenReview forum: "Permutation-Invariant Hierarchical Representation Learning for Reinforcement-Guided Feature Transformation"
_ICLR.cc/2026/Conference — Submitted to ICLR 2026_

### Official Review · Reviewer_upia · 2025-10-27

**Soundness:** 2
**Presentation:** 1
**Contribution:** 2
**Rating:** 2
**Confidence:** 3

**Summary:**

This paper proposes a novel tabular feature transformation method aimed at addressing three limitations of existing works. First, two features at different hierarchies are designed to learn token-level and concept-level representations, thereby addressing the first limitation of ignoring the hierarchical relationships among the original features. Second, a specific self-attention pooling mechanism is adopted to learn permutation-invariant features, thus overcoming the second limitation of neglecting order invariance in the generated features. Thirdly, Proximal Policy Optimization is used to avoid the drawbacks of traditional gradient-based optimization methods. Experiments are conducted on various datasets in comparison with a rich set of baseline methods.

**Strengths:**

(1) The motivations and descriptions of the existing limitations are well introduced and clearly explained, especially the examples presented in Figure 1.

(2) The quantitative results in Table 1 and the ablation studies in Figure 3 show that the proposed method is promising, and each design module contributes effectively.

**Weaknesses:**

(W1) The major weakness of this paper is the writing and organization. For some reason, the authors break the descriptions of the pipeline of the proposed methods into tiny fragments scattered throughout the methods section, which makes it difficult to follow and understand both the overall pipeline and the details of the proposed methods. Instead of writing several pieces of "why something matters" and "why use something," please write the entire problem statement and the proposed method into concise and coherent paragraphs using consistent notations and terms.

For example, (1) The term “aggregation layer” used in Figure 2 and line 227 is never formally defined, but it appears to be a mean pooling layer as described in line 201, which takes readers extra efforts to link them. (2) The important token embedding notation E is not mentioned in the summarized introduction of the problem statement, while other key terms such as Γ and G are mentioned. (3) The relationship between E and G is unclear: while the authors provide many vague plain-language explanations, a clear and formal equation for their relationship is not found. Unfortunately, the closest one seems to contain typos in its definition: In line 229, the authors write “Thereafter, G′ is input into φ_con to obtain the global embedding G′ = φ_con(Mean(E)).” But it looks like the first G′ should be G instead, given the text before this line? Additionally, the “suggested” definition here G = Mean(E) seems inconsistent with the examples provided in line 206, where G is the mean over a set of E.

In summary, similar readability issues make it difficult to understand and evaluate the proposed method, and a simple step-by-step explanation corresponding to Figure 2, using formal languages and equations, is needed.

(W2) While the performance of the features transformed by the proposed method is evaluated on downstream tasks, no direct evaluation was conducted on the embeddings themselves. For example, although the authors claim that “This structure guarantees that any permutation of generated concepts yields identical embeddings” in line 96, no direct evidence supports this claim for the dataset used in this paper’s experiments.

**Questions:**

(Q1) The authors are encouraged to respond to and explain my concerns listed in the Weaknesses section. Additional experimental results and demonstrations would also be greatly appreciated.

(Q2) It appears that the URL to the paper’s code is broken. I received a “The requested file is not found” error for all .py files in the code folder.

---

> ### Author Response · Authors · 2025-11-20
> **Authors' Rebuttal**
>
> Dear Reviewer upia,
>
> Thank you for your positive assessment of our paper’s motivation and the promising empirical performance demonstrated in our experiments.
>
>
>
> **1. Regarding Writing and Organization**
>
> - Our original intention was to avoid overloading the notation in the main paper. Since the mean-pooling operation that forms concept embeddings was already described earlier (line 201), we referred to this process as the “aggregation layer” in Figure 2 for brevity, assuming that readers would connect it back to the previously introduced mean-pooling step.
>
>   **However**, we notice that the name “aggregation layer” was mentioned in the figure before being explicitly connected to the mean-pooling operation in the text. We have now explicitly stated in the encoder description that the mean-pooling step *serves as the aggregation layer*, thereby unifying terminology across Figure 2, the method description, and the implementation.
>
> - The problem statement was originally designed to emphasize the *global embedding space* $\mathbb{G}$,  which is the core object optimized by our policy-guided search. Because $\mathbf{E}$ and $\mathbf{G}$ are intermediate representations (token-level and concept-level respectively), we intentionally avoided introducing too many low-level symbols in the high-level summary to maintain conceptual focus and narrative flow.
>
>   In response to the reviewer's suggestion, we have added concise definitions of token embeddings $\mathbf{E}$ and concept embeddings $\mathbf{G}$ directly in the problem statement section.
>
> - The conceptual embedding $\mathbf{G}$ is computed by mean-pooling token embeddings that belong to the same concept. In the original manuscript, because the definition of $\mathbf{G}$ had already been provided the expression $\phi_{con}(\text{Mean}(E))$ was intended as a shorthand reference to this pooling operation rather than a redefinition of the global mean.
>
>   We now consistently use $\mathbf{G'} = \phi_{con}(\mathbf{G})$. All ambiguous shorthand expressions have been removed.
>
> **2. Regarding Permutation Invariance**
>
> - Our concept encoder is built upon the Pooling by Multihead Attention (PMA), which is the core module of the Set Transformer architecture [1]. The permutation-invariance property of PMA has been formally established in the original Set Transformer work [1], which provides the theoretical guarantee for our architecture.
>
>   To further verify this property, we include an empirical study on permutation sensitivity. For each example, we randomly shuffle the order of the concept embeddings several times and input both the original and permuted versions to the concept encoder. We then visualize the resulting global embeddings. The permuted versions consistently cluster around the original one, showing that the encoded representation is stable under reordering. This provides empirical support for the permutation invariance guaranteed by the PMA module.
>
>   Lee, J., Lee, Y., Kim, J., Kosiorek, A., Choi, S., & Teh, Y.W. (2019). Set Transformer: A Framework for Attention-based Permutation-Invariant Neural Networks. Proceedings of the 36th International Conference on Machine Learning (PMLR 97), 3744–3753. Available at https://proceedings.mlr.press/v97/lee19d.html.
>
> **3. Regarding Code Availability**
>
> - The code was hosted on the anonymous 4open platform as required for anonymized submissions. It appears that the platform occasionally returns a “file not found” error depending on the access route or temporary availability. To ensure reliable access, we have re-checked and re-uploaded the complete code in the Supplementary Material of the revised submission.
>
> Thank you again for your constructive feedback. We believe the revisions have substantially improved the clarity and completeness of the paper.

---

### Official Review · Reviewer_bviC · 2025-10-31

**Soundness:** 3
**Presentation:** 3
**Contribution:** 2
**Rating:** 4
**Confidence:** 4

**Summary:**

This paper solves the problem of automated feature transformation using hierarchical modeling and policy-guided feature transformation. The proposed framework first maps the feature transformation sequence into the continuous embedding space and then employs the policy-guided multi-objective search to find the optimal feature transformation sequence in the continuous embedding space. Experimental results reveal the effectiveness of the propose method.

**Strengths:**

1. The paper is well-written and easy to follow.

2. The proposed hierarchical encoder-decoder models could generate better global embeddings.

3. The reward model in the policy-guided search contains the length of feature transformation sequence, which is helpful to find more compact transformation sequence and improve computation efficiency.

4. The extensive experiments demonstrate the superior performance of the proposed method.

**Weaknesses:**

1. The novelty is limited. The overall framework that first maps the feature transformation sequence into the continuous embedding space and then perform optimization in the continuous space is not new and is prosed in many existing methods such as DIFER Zhu et al. (2022b) and MOAT Wang et al. (2023). Thus, the innovation is incremental.

2. The proposed method is complicated and the involve huge computation. For example, in the hierarchical modeling stage, token-level encoder/decoder, concept-level encoder/decoder need to be trained. In the policy-guided multi-objective search stage, PPO agent needs to be trained. Thus, the computation complexity should be analyzed in detail.

3. Beside the performance comparison, the computational efficiency comparison should be performed, especially on large-scale datasets.

4. It is not clear why the proposed self-attention pooling mechanism can ensure permutation invariance. The authors should further explain the permutation invariance in detail and demonstrate permutation invariance with some case studies.

5. It is suggested to analyze the interpretability of the generated features.

**Questions:**

see Weakness.

---

> ### Author Response · Authors · 2025-11-20
> **Authors' Rebuttal**
>
> Dear Reviewer bviC,
>
> Thank you for your positive feedback on our paper’s writing quality, model design, and experimental results.
>
> **1. Regarding Novelty**
>
> The contributions of PHER are three-fold:
>
> - Hierarchical modeling beyond token-level representations: Previous methods typically only encode feature transformations at token level. In contrast, introduces a **hierarchical encoder–decoder framework** that models feature–operator interactions at the token level and abstracts them into concept-level representations before forming a global embedding. This multi-level structure enables richer semantic understanding of generated features and has not been considered in existing works.
> - Permutation-invariant global embedding space: Existing approaches encode transformation sequences using sequential models (e.g., RNN/Transformer-based), which makes the learned embeddings sensitive to the arbitrary order of generated concepts. In contrast, PHER adopts a **permutation-invariant mechanism** that removes this order dependency and produces stable global embeddings.
>
> - Policy-guided search in the non-convex embedding space: Prior methods generally rely on **gradient-based optimization** in the embedding space, which can be unstable or ineffective in non-convex spaces. PHER instead adopts a **policy-guided search strategy** that directly interacts with the permutation-invariant global embedding space, enabling more stable reward-driven exploration.
>
> **2. Regarding Computational Efficiency**
>
> - We have already analyzed time–complexity of PHER in **Appendix D.2**, where we examine how PHER’s search time and inference time scale with feature dimensionality and sample size across multiple datasets.
>
> - Moreover, we evaluate PHER on two high-dimensional, large-scale datasets and report both training time and inference time in comparison to the SOTA model MOAT. The results show that PHER maintains competitive computational efficiency while achieving superior predictive performance.
>
> - Baron_mouse: **1886** samples, **14878** features
>
>   | **Model** | **Training Time (s)** | **Inference Time (s)** | **Performance** |
>   | :-------- | :-------------------- | :--------------------- | :-------------- |
>   | PHER      | 3512.58               | 21.4736                | 0.9720          |
>   | MOAT      | 3340.73               | 14.1132                | 0.9668          |
>
> - Baron_human : <b>8569</b> samples, <b>20126</b> features
>
>   | **Model** | **Training Time (s)** | **Inference Time (s)** | **Performance** |
>   | :-------- | :-------------------- | :--------------------- | :-------------- |
>   | PHER      | 4959.42               | 22.9195                | 0.9631          |
>   | MOAT      | 4750.85               | 15.1665                | 0.9578          |
>
> **3. Regarding Permutation Invariance**
>
> - The permutation invariance in PHER comes from the **Pooling by Multihead Attention (PMA)** module used in our concept encoder, which follows the Set Transformer architecture [1]. PMA is mathematically proven to be **invariant to any permutation of its input elements**, providing the theoretical foundation for our design.
>
> - To further verify this property, we also include a **permutation-sensitivity study** in our paper **(Appendix D.4)**. In this experiment, we randomly change the order of concept embeddings for multiple samples and visualize the resulting global embeddings. Across all datasets, the permuted versions cluster tightly around the original ones, demonstrating that PHER produces **stable global embeddings regardless of input concept order**.
>
>   Lee, J., Lee, Y., Kim, J., Kosiorek, A., Choi, S., & Teh, Y.W. (2019). Set Transformer: A Framework for Attention-based Permutation-Invariant Neural Networks. Proceedings of the 36th International Conference on Machine Learning (PMLR 97), 3744–3753. Available at https://proceedings.mlr.press/v97/lee19d.html.
>
> **4. Regarding Interpretability of Generated Features**
>
> - We would like to clarify that PHER’s interpretability has already been analyzed in **Appendix D.3**, where conduct a traceability case study showing how PHER identifies meaningful features (e.g., *alcohol* in Wine Quality) and generates semantically coherent transformed concepts that align with domain knowledge.

---

### Official Review · Reviewer_3iT1 · 2025-10-31

**Soundness:** 2
**Presentation:** 2
**Contribution:** 1
**Rating:** 2
**Confidence:** 4

**Summary:**

This work proposes PHER to automatically transform tabular features. The core ideas are permutation-invariant hierarchical modeling and policy-guided multi-objective search. The former models both low-level nad high-level features interactions, using a self-attention pooling mechanism, while the later uses reinforcement learning to explore non-convex embedding space.

**Strengths:**

- Automatic feature engineering is an interesting topic.
- The empirical result is strong and shows the superior performance of PHER.
- This work provides source code for reproducibility.

**Weaknesses:**

- The RL approach might be computationally expensive, expecially on large datasets and high dimenstional datasets.
- Automatic feature engineering is an old topic. The RL approach of automatic feature engineering has been studied in 2017 [1].

[1] Nargesian, Fatemeh, et al. "Learning feature engineering for classification." Ijcai. Vol. 17. 2017.

**Questions:**

- For table 1, could you provide a supervised learning baseline without any feature engineering?
- Can a model trained on one dataset (e.g. Higgs Boson) be applied on another dataset (e.g. Amazon Employe)?

---

> ### Author Response · Authors · 2025-11-20
> **Authors' Rebuttal**
>
> Dear Reviewer 3iT1
>
> Thank you for your positive feedback on the strong empirical results and the reproducibility of our work.
>
> **1. Regarding Novelty**
>
> We acknowledge that LFE (2017) explored the use of learning-based methods for recommending feature transformations. Earlier methods operate in the *symbolic transformation space*, predicting which predefined transformations should be applied to individual features. They do not model token-level interactions, do not learn concept abstractions, and do not construct a continuous embedding space. However, PHER is fundamentally different from these prior approaches.
>
> In PHER, we introduce:
>
> - **A hierarchical encoder–decoder framework** that models transformations at both token level and concept level, enabling abstraction beyond symbolic search.
> - **A permutation-invariant search space**, which was not present in early RL approaches and fundamentally changes how feature transformations are represented.
> - **Policy-guided search in the non-convex embedding space**, rather than searching over symbolic actions, allowing more stable and efficient exploration than discrete RL policies used in earlier work.
>
> **2. Regarding Computational Efficiency**
>
> - The time–complexity of PHER has already been analyzed in **Appendix D.2**, where we study how both search time and inference time vary with feature dimensionality and sample size across several datasets.
>
> - In addition, we further assess PHER on two large, high-dimensional datasets and compare its runtime with the SOTA method MOAT. Despite PHER’s richer modeling components, its training and inference times remain in a comparable range while yielding better predictive performance.
>
> - Baron_mouse: **1886** samples, **14878** features
>
>   | **Model** | **Training Time (s)** | **Inference Time (s)** | **Performance** |
>   | :-------- | :-------------------- | :--------------------- | :-------------- |
>   | PHER      | 3512.58               | 21.4736                | 0.9720          |
>   | MOAT      | 3340.73               | 14.1132                | 0.9668          |
>
> - Baron_human : **8569** samples, **20126** features
>
>   | **Model** | **Training Time (s)** | **Inference Time (s)** | **Performance** |
>   | :-------- | :-------------------- | :--------------------- | :-------------- |
>   | PHER      | 4959.42               | 22.9195                | 0.9631          |
>   | MOAT      | 4750.85               | 15.1665                | 0.9578          |
>
> **3. Regarding Performance without Feature Engineering**
>
> - We have added a supervised learning baseline that trains a Rdirectly on the original feature space without applying any feature transformations. The results have already been incorporated into the main results table and are also shown below for clarity.
>
>   |      Dataset       | Original Performance |  PHER  |
>   | :----------------: | :------------------: | :----: |
>   |    Higgs Boson     |        0.6957        | 0.7074 |
>   |  Amazon Employee   |        0.9302        | 0.9361 |
>   |     PimaIndian     |        0.7756        | 0.8326 |
>   |       SpectF       |        0.7596        | 0.9291 |
>   |     SVMGuide3      |        0.7781        | 0.8524 |
>   |   German Credit    |        0.6488        | 0.7356 |
>   |   Credit Default   |        0.8019        | 0.8103 |
>   | Messidor_features  |        0.6329        | 0.7442 |
>   |  Wine Quality Red  |        0.4562        | 0.5635 |
>   | Wine Quality White |        0.5364        | 0.5562 |
>   |      SpamBase      |        0.9268        | 0.9357 |
>   |  AP-omentum-ovary  |        0.8449        | 0.8832 |
>   |    Lymphography    |        0.1411        | 0.5333 |
>   |   MNIST fashion    |        0.7119        | 0.7355 |
>   |   Housing Boston   |        0.4145        | 0.5445 |
>   |      Airfoil       |        0.5189        | 0.6459 |
>   |     Openml_589     |        0.5095        | 0.6694 |
>   |     Openml_618     |        0.4689        | 0.6925 |
>   |     Openml_620     |        0.5101        | 0.6517 |
>
> **4. Regarding Transferability across Datasets**
>
> - Although PHER learns generalizable representations, the generated concepts are still dataset-specific. Because datasets differ in feature semantics, distributions, and prediction targets, directly applying a model trained on one dataset (e.g., Higgs Boson) to a different dataset (e.g., Amazon Employee) is not feasible without re-training or adaptation.
> - This is consistent with prior work in feature engineering and tabular representation learning, where **cross-dataset transfer remains an open challenge**. As an extension, we consider developing a *feature-transformation foundation model* capable of generalizing across diverse datasets as a promising future direction.

---

### Official Review · Reviewer_gQqC · 2025-10-31

**Soundness:** 2
**Presentation:** 4
**Contribution:** 3
**Rating:** 4
**Confidence:** 3

**Summary:**

This paper proposes PHER, a unifying framework for automated feature transformation on tabular data, where the authors try to tackle the three main challenges of recent generative methods using the three stages, specifically by modeling hierarchical features, introducing permutation-invariant feature sets, and using a non-gradient approach (PPO) for final search.

I'll adjust my rating based on the author's response.

**Strengths:**

1. Well-written paper, with visually appealing figures.
2. The proposed system (using a permutation-invariant mechanism on the set of generated concepts) is technically sound.
3. Strong empirical results in quantitative performance.

**Weaknesses:**

1. The proposed system seems to have a complex design. The workflow requires 1) initial data collection using reinforcement learning; 2) a 3-stage procedure for training the hierarchical model; and 3) a final PPO search. This could introduce a barrier to adoption.
2. Fig. 5 shows that using high-quality seeds is much better than using random seeds. The paper does not fully explore how PHER would perform if the initial data collection phase yielded a weaker set of records. We do not know what the lower bound of quality is. Also, this shows a dependence on the random seed, which renders the proposed solution less robust.

**Questions:**

1. The mean pooling layer is illustrated with an example: $G_1 = \text{Mean} (E_{f_1}, E_{plus}, E_{f_7})$ for the transformation $f_1 + f_7$, which seems to be at risk of losing the crucial syntax structure of the transformation. For instance, how this simple mean pooling can distinguish between $f_1 + f_7$ and $f_7 \times f_1$ (the operator token is just one part of the average)? Is this a potential information bottleneck here?
2. How dependent is the method on the quality / diversity of the initial set of transformation records (which is used to train the hierarchical model and provide the top-$k$ for the search)? An ablation study on this would be helpful.
3. The PPO agent introduces several hyperparameters. Fig. 7 shows an optimal $\lambda$ of $0.9$, which means it prioritizes performance over the stated goal of minimizing transformation length. However, from Fig. 6, PHER does still produce shorter sequences. How does this justify the emphasis on the multi-objective balance? Also, given the complexity of the system, tuning these hyperparameters could be challenging in practice.


Also, I am not able to see the content of the files in the link provided, except for the `README.md`. I can only see the file structure of the repo. I cannot determine how complex the implementation is from the code.

---

> ### Author Response · Authors · 2025-11-20
> **Authors' Rebuttal**
>
> Dear Reviewer gQqC,
>
> Thank you for your positive assessment of our paper’s clarity, technical soundness, and empirical results.
>
> **1. Regarding Model Complexity**
>
> - The initial RL data collection is a one-time offline process. It is only used to build a dataset of transformation sequences. Once the dataset is prepared, users do *not* need to run RL again.
> - The three-stage training is fully automated. The three steps can be executed by a single training script. From the user's perspective, this is equivalent to **training a standard encoder–decoder model**, without any extra manual intervention.
> - The final PPO search employs a very lightweight policy. After training the hierarchical model, the PPO agent only operates in the low-dimensional space. This significantly reduces search cost and results in more efficient and stable search than gradient-based search methods.
> - Our ablation study (Figure 3) show that removing any stage results in consistent performance drops. We therefore believe the multi-stage structure is justified by empirical evidence.
>
> **2. Regarding Seeds Quality**
>
> - As shown in Fig. 5, high-quality seeds accelerate the search, however, **random seeds still lead to stable exploration and competitive final performance.** Even with weak seeds, the PPO agent continues to update the search direction through feedback, maintaining stable progress. Thus, the dependence is limited and does not undermine robustness.
> - Moreover, since PPO naturally balances exploration and exploitation, the agent is not restricted by the initial seeds. If the seed quality is low, the agent can simply rely on **stronger exploration** by adjusting RL hyperparameters such as increasing the entropy bonus or loosening the clipping range. These common settings encourage broader search and help the agent move toward higher-reward regions as feedback accumulates.
>
> - The hierarchical model is **also robust to noisy or suboptimal seeds.** It is trained on the entire distribution of RL-collected transformation records, which already contains trajectories of varying quality. Through the self-attention layers in the hierarchical encoders, these records are abstracted into stable high-level representations. As a result, meaningful patterns are emphasized, while less informative ones are down-weighted. This prevents low-quality seeds from dominating the learned hierarchy.
>
> **3. Regarding Mean Pooling Layer Preservation**
>
> - In PHER, the mean pooling layer is applied only *after* the token-level encoder has already processed the sequence and embedded the operator semantics and local syntax into contextualized token representations. By the time pooling is performed, the embeddings of $f_1$, $+$, and $f_7$ already encode their functional interactions, which differentiates expression such as $f_1 + f_7$ and $f_7 \times f_1$. Thus, **the structural information is already preserved at the token level.** The mean pooling does not play a role in encoding or distinguishing syntax.
>
> **4. Regarding PPO Hyperparameters**
>
> - Although Fig. 7 shows that the best λ is 0.9, this does not conflict with the shorter sequences in Fig. 6. The value of λ only determines how PPO balances performance improvement and the length penalty in the reward; it does not force the model to use more or fewer operations. During concept-level search, **PPO emphasizes concepts that meaningfully improve downstream performance**, while concepts that contribute little or introduce noise are naturally down-weighted. When decoding, these **weak concepts tend to produce padding or uninformative steps**, while only the useful concepts become actual transformation operations. As a result, the **final transformation contains fewer effective steps**, leading to cleaner and shorter transformations even when λ prioritizes accuracy.

---

> > ### Comment · Reviewer_gQqC · 2025-11-27
> >
> > Regarding the claim that the model is "robust to suboptimal seeds". Is there empirical evidence to back this up? The rebuttal suggests that users can simply "adjust RL hyperparameters" (like entropy bonuses) to fix low-quality seeds. First, how does the user know it is encountering low-quality seeds if it was being "trained as a standard encoder–decoder model, without any extra manual intervention"? Second, RL hyperparameters are sometimes finicky. Is there any evidence to show its robustness or any experiment when starting with a suboptimal seed?
> >
> > "these weak concepts tend to produce padding or uninformative steps" is an interesting fact and leads to the non-common-sensical observations highlighted in my original comment. Are there any qualitative analyses? How about if we do not even consider the length in the objective at all?

---

### Meta-Review · Area_Chair_iNBA · 2026-01-06

**Summary:**

This paper addresses an interesting problem; however, the proposed model design is overly complex and incurs substantial computational cost. Although the method operates in an offline setting, it requires recomputation when applied to a new benchmark. Moreover, the reported performance gains are marginal and lack comprehensive experimental validation. As a result, the reviewers provided negative assessments.

**Reviewer Concerns:**

The computation cost and novelty of the model design.

**Reviewer Scores:**

If the model computation can be addressed well, they may raise the score, but it is hard.

---

### Decision · Program_Chairs · 2026-01-26

Reject